# Safety and Efficacy of Daptomycin in Neonates with Coagulase-Negative Staphylococci: Case Series Analysis

**DOI:** 10.3390/antibiotics10020168

**Published:** 2021-02-07

**Authors:** Yahya Mohzari, Fahad Aljobair, Ahmed Alrashed, Syed Mohammed Basheeruddin Asdaq, Renad Abdullah Alshuraim, Suzan Suhail Asfour, Mountasser Mohammad Al-Mouqdad, Reem F. Bamogaddam, Deemah Al-Anazi, Catherine E. Zeilinger, Ahmad Alamer, Batool Mohammed Alhassan, Nagaraja Sreeharsha

**Affiliations:** 1Clinical Pharmacy Department, King Saud Medical City, Riyadh 12746, Saudi Arabia; Yali2016@hotmail.com (Y.M.); r.alshuraim@ksmc.med.sa (R.A.A.); asfsuzan@gmail.com (S.S.A.); reem1414faisal@gmail.com (R.F.B.); 2Pediatric Infectious Disease, King Saud Medical City, Riyadh 12746, Saudi Arabia; fahad@ksmc.med.sa; 3Pharmaceutical Services Administration, Inpatient Department, Main Hospital, KFMC, Riyadh 11564, Saudi Arabia; emadasdaq@gmail.com; 4Department of Pharmacy Practice, College of Pharmacy, AlMaarefa University, Dariyah 13713, Riyadh, Saudi Arabia; 5Neonatal Intensive Care Unit, King Saud Medical City, Riyadh 12746, Saudi Arabia; m.almouqdad@ksmc.med.sa; 6Inpateint Pharmacy Department, King Saud Medical City, Riyadh 12746, Saudi Arabia; deemah@ksmc.med.sa; 7Pharmaceutical Service Department, Clinical Pharmacy Section, King Fahd Medical City, Riyadh 11564, Saudi Arabia; qualityasdaq@gmail.com; 8Center for Health Outcomes and PharmacoEconomic Research, University of Arizona, Tucson, AZ 85721, USA; alamer@pharmacy.arizona.edu; 9Department of Clinical Pharmacy, Prince Sattam Bin Abdulaziz University, Alkharj 16273, Saudi Arabia; 10Neonatal Intensive Care Unit, AlMoosa Specialist Hospital, Alahsa 36342, Saudi Arabia; batool42@gmail.com; 11Department of Pharmaceutical Sciences, College of Clinical Pharmacy, King Faisal University, Al-Ahsa 31982, Saudi Arabia; sharsha@kfu.edu.sa; 12Department of Pharmaceutics, Vidya Siri College of Pharmacy, Off Sarjapura Road, Bangalore 560035, India

**Keywords:** daptomycin, neonates, *Coagulase-negative staphylococci*, retrospective, safety, efficacy

## Abstract

There has been an increase in the prevalence of gram-positive bacteremia in neonates in the last two decades. However, as a consequence of better care, there has been an increase in the survival of premature neonates. *Coagulase-negative staphylococci* (CoNS) is the most prevalent bacteria, responsible for up to 60% of late-onset sepsis (LOS). Daptomycin, a lipopeptide antimicrobial agent, is active against CoNS. This was an observational, retrospective case series study carried out in the Pediatric Hospital of King Saud Medical City, Riyadh, Saudi Arabia. The medical records of 21 neonates, aged 0–28 days, who were treated in Neonatal Intensive Care Unit (NICU) with intravenous daptomycin as monotherapy or combination therapy for at least 4 days for proven gram-positive infection between June 2019 to July 2020, were included. The median gestational and chronological age were 27 weeks and 5 days, respectively. The most frequent diagnosis in neonates was infective endocarditis (42.9%). Of the 21 patients who received daptomycin therapy, 13 (62%) recovered and 8 died. The clinical cure rate was higher in *Staphylococcus hominis* (100%) and in patients who received 6 mg/kg/dose twice daily (62.5%). The mean of aspartate aminotransferase significantly elevated after starting daptomycin (*p* = 0.048). However, no muscular or neurological toxicity of daptomycin was documented in any of the cases. Overall, daptomycin was well tolerated, even with long-term treatment.

## 1. Introduction

Eli Lilly and Company (Lilly) discovered daptomycin in the early 1980s. Later, during the late 1980s and early 1990s, Lilly designed clinical studies of intravenous daptomycin. The outcome of trials was significant in treating skin and soft tissue infections and bacteremia. Later, Cubist Pharmaceuticals Inc. was licensed (1997), with global rights to develop daptomycin for the treatment of serious gram-positive infections for inpatients. Cubist conducted multiple clinical trials in the USA and Europe in 1999 [1,2]. Subsequently, the US Food and Drug Administration (FDA) approved daptomycin for complicated skin and skin structure infections (cSSSI) both for adult and pediatric patients (<17 years) and *Staphylococcus aureus* (SA) bacteremia (including right-sided endocarditis) in adult patients [3].

Daptomycin has been reported to possess bactericidal and dose-dependent activity with a low potentiality to develop resistance. Daptomycin works on the bacterial membrane, causing rapid depolarization by inhibiting the intracellular synthesis of DNA, RNA, and protein, resulting in bacterial cell death [4]. Daptomycin showed excellent overall activity against indicated species (including *Staphylococci*, *Enterococci*, and *Streptococci*) in several geographic regions. It was also active against *Coagulase-Negative Staphylococci* (CoNS), including vancomycin-non-susceptible isolates [5].

The prevalence of gram-positive bacteremia in infants and neonates has increased in the past two decades. Around 90% of hospitalized infants of late-onset sepsis (LOS) infected with *staphylococci* species. CoNS is the most prevalent bacteria, causing up to 60% of LOS. The majority of the CoNS isolates are *S. epidermidis* (>80%), followed by *S. capitis* (13%), *S. hominis* (2%), *S. warneri* (2%), and *S. haemolyticus* (1%). 

Antibiotic resistance in CoNS, especially against beta-lactam antibiotics, has surged in the last few years. Studies have revealed that more than 90% of the CoNS isolates causing late-onset sepsis in neonates are methicillin-resistant isolates (MR-CoNS). Moreover, >50% of these CoNS isolates have resistance to more than three classes of antibiotics. However, the high prevalence of methicillin-resistant CoNS in NICUs usually leads to frequent vancomycin use. Recent studies have raised concerns about the spread of vancomycin-thermoresistant *S. capitis* strains and their involvement in persistent bacteremia despite prolonged vancomycin therapy [6,7,8,9,10,11,12,13]. In addition, studies have found poor clinical outcomes related to heterogeneous vancomycin-intermediate *S. aureus* and vancomycin-intermediate *S. aureus* infections, and limited treatment options for *S. aureus* infections with reduced susceptibility to vancomycin [14,15] have warranted the need to explore viable alternative antimicrobial agents with enhanced activity against MR-CoNS. Further, a study on the use of daptomycin in animal models with MR-CoNS showed greater bactericidal activity, along with a quicker onset of action, prolonged half-life, and reduced inflammatory reaction [16]. More recently, the successful use of daptomycin for treatment of MR-CoNS has been reported in number of clinical studies [17,18]. 

In the current resistance pattern with the very limited options in neonates, treating neonatal CoNS infection has become challenging. Unfortunately, there is no formal evaluation of daptomycin in infants, and the manufacturer of daptomycin recommends avoiding using daptomycin in patients <12 months due to musculoskeletal, neuromuscular, and nervous system adverse effects. Daptomycin is identified in the Key Potentially Inappropriate Drugs in Pediatrics (KIDs) list due to the risk of neuromuscular and skeletal adverse events [19]. These adverse events (including twitching, muscle rigidity of the limbs, and impaired use of the limbs) were reported in the preclinical studies on neonatal animal models (canine) [20]. These neonatal adverse events occurred with canine doses and drug exposure levels that were higher than the standard adult human dose (6 mg/kg) and resolved within 28 days of discontinuation [21]. 

Although daptomycin is included in KIDs list, it has weak evidence on its neonatal toxicity, and there is balance of risk to weigh between “unknown risk in neonate of a very efficient antibiotic in adults” and “severity of CoNS infection in premature neonates”. Hence, in our institution, daptomycin was considered as second-line therapy after vancomycin treatment failure, defined as persistent bacterial growth on top of vancomycin treatment and lack of satisfying clinical response. Therefore, the purpose of this observational study was to analyze the safety and efficacy of daptomycin in neonates with CoNS infections.

## 2. Results

### 2.1. Patients’ Characteristics

A total of 21 neonates were identified who received daptomycin upon hospitalization. The baseline characteristics of the study patients are shown in Table 1. The median (range) gestational age was 27 weeks (24–37), and the median (range) chronological age was 5 days (2–26). The median weight (range) was 870 grams (530–3700). Of the patients, 11 (52.4%) were female, and 20 (95.2%) were pre-term. The most frequent diagnosis in a neonate was infective endocarditis, which was found in nine of patients (42.9%). The common causative microorganism was *Staphylococcus epidermidis*, which was found in 15 of patients (71.4%). All microorganisms were sensitive to vancomycin and resistant to cloxacillin. In all cases, daptomycin was administered as second-line therapy after vancomycin treatment failure, which is defined as persistent bacterial growth on top of vancomycin treatment (15 mg/kg, twice daily for 5 days, *iv*) and lack of satisfying clinical response. Continuous infusion of vancomycin was an option in the absence of desired therapeutic response of its twice daily dosing. However, for patients at high risk for renal impairment, such as the preterm neonates in this study, alternatives must be considered. Most of the neonates (20 out of 21, 95.2%) in this study were preterm and possibly had a decreased number of nephrons than full-term birth neonates. Therefore, availing an alternative agent such as daptomycin was considered. Most (20 out of 21) of the neonatal patients in the study received monotherapy daptomycin (95.2% of patients), with the 10 mg/kg/once dosing given to 10 out of the total 21 patients (47.6%). The median (range) duration of daptomycin therapy was 22 (4–43) days and the median (range) follow-up duration was 82.5 (37–176) days. The baseline and on-treatment recording of parameters such as white blood cells (WBC), platelets, kidney, and liver functions are given in the Appendix A.

### 2.2. Outcomes

During hospitalization, out of the 21 patients who received daptomycin, 13 patients (61.9%) were cured of the infection and 8 (38.1%) died. The most relevant 3 of death were gram-positive infection (4 patients, 50.0%), gram-negative infection (three patients, 37.5%), and abdominal perforation (1 patient, 12.5%). Table 2 provides a more detailed overview of the dosing, duration, and clinical outcome for each diagnosis. In terms of outcomes associated with causative microorganisms, the number of cases with clinical cure was high with *Staphylococcus epidermidis* (73.3%, 11 out of 15 cases). Given the diagnosis-based response, infective endocarditis causes had more deaths (44.1%) than any other diagnoses. The cure and death rates related to the dosing regimen were varied. The dosing of 10 mg/kg/dose once daily produced a similar number (five each) of deaths and recoveries. Out of the 15 cases of *Staphylococcus epidermidis*, 7 of them received 10 mg/kg dosing and 4 had clinical cure (57.14%), while the dosing of 6 mg/kg in this group of patients (*Staphylococcus epidermidis* infected) produced a cure rate of 80%. Overall, the clinical cure rate was higher than the death rate among patients who received 6 mg/kg/dose twice daily with a proportion of 62.5% (5 out of 8) and 37.5% (3 out of 8), respectively.

### 2.3. Adverse Effect

The findings demonstrated that the mean of liver enzymes (only aspartate aminotransferase) significantly elevated after starting daptomycin compared with the mean of liver enzyme baseline (baseline = 86.1 and post-therapy = 159, *p* = 0.048). Two patients developed mild rashes related to daptomycin, which did not necessitate discontinuing the daptomycin. No significant elevation in creatine phosphokinase (CPK) readings were noted. Moreover, providers’ notes did not mention any muscular or neurological toxicity of daptomycin.

## 3. Discussion

Antimicrobial resistance is a serious problem in the treatment of microbial infection. One of the highly potent antibiotics that targets gram-positive bacteria is daptomycin. It has some characteristics, such as greater bactericidal activity with quicker onset of action, longer half-life in the ventricles of the brain, and significantly low inflammatory reaction in the host, that favor its use in the treatment of persistent bacterial growth. In addition to this, the efficacy of vancomycin is directly linked with its trough plasma concentration. In susceptible individuals, such as those with reduced renal function or incomplete formation of renal system, including the case of preterm neonates of this study, increasing the dose of vancomycin beyond therapeutic dose is associated with nephrotoxicity. Therefore, persistent bacterial growth at the end of 5 days of vancomycin therapy poses a need for a potent antimicrobial with enhanced activity against MR-CoNS. Although the use of daptomycin is restricted in neonates, it has shown a good cure rate in several studies [22,23]. Our study included a case series of 21 newborns with Staphylococci infections, and we found that daptomycin intervention provided successful recovery in around 62% of the patients. Despite having a lower cure rate compared to the earlier report [24], the outcome of the study is significant, as most of the patients received daptomycin monotherapy (95%) and had premature birth (95%).

The dose of daptomycin is critical for its therapeutic benefits in all groups, particularly in the neonate. Although, both FDA in the USA [25] and EMA in Europe [26] have approved the use of daptomycin for the treatment of adult patients with skin and soft tissue infections and several *Staphylococcus aureus* infections, its pediatric use is still prohibited due to the non-existence of data on safe dosage. However, due to its greater importance in the treatment of several severe infections in pediatric patients, there are attempts to regularize its dosage. Overall, in most of the cases, we studied in this research, a dose of 6 mg/kg twice daily produced better results compared to 10 mg/kg once daily. This findings were consistent across different causative organisms of the clinical cases included in this research. This indicate that the variation in the outcomes is based on the antibiotic regimen rather the strain of the causative organisms studied. Our findings are similar to the earlier published reports [27,28]. The differences in the behavior of different doses of daptomycin are possibly due to the involvement of pharmacokinetic variables. Protein binding of daptomycin is responsible for a relatively long duration of pharmacological effect, whereas, absence or lack of protein binding sites for daptomycin in neonate result in its faster clearance [29,30]. Therefore, 10 mg/kg once daily gets cleared from the plasma with decreased duration of action and lesser therapeutic benefits compared to a low dose (6 mg/kg) of daptomycin administered twice daily. 

Our study also showed infective endocarditis (IE) with the highest failure rate. The high lethality with IE might be due to the rapid progression of infection to intractable congestive heart failure before the onset of daptomycin therapy. Therefore, delaying the introduction of daptomycin has to be re-examined to secure enhanced antibacterial benefits.

Most of the cases included in this study exhibited elevation in liver enzyme levels due to daptomycin administration compared to the baseline readings. Unlike other studies [31], no significant increase was noticed in CPK in any included case. Nevertheless, this elevation is expected to reverse upon discontinuation of therapy. With this exception, long -erm use of daptomycin has not shown any other significant adverse effects, which is similar to earlier findings [32].

One of the major limitations of this study is its retrospective nature. We also recognize that the selection of the patients was nonrandomized and heterogeneous. There were variations in the type of infections and daptomycin dosage among different cases. Further, data on muscular or neurological toxicity of daptomycin were missing, which could have been due to either the absence of such adverse effects or a possible miss by an observer. However, the researchers of this study are working to trace the surviving children for re-examining to detect potential neuromuscular damage with the help of neuropediatric specialists within 1 year from the time of administration of daptomycin.

Although the outcome of the study showed better efficacy for twice daily dosing of daptomycin at 6 mg/kg, further investigations are warranted to check the relevant pharmacokinetic parameter and to determine their correlation with the therapeutic outcomes.

## 4. Materials and Methods

### 4.1. Design, Setting, Patient Selection

This was an observational, retrospective case series study carried out in the Pediatric Hospital, one of the leading hospitals in King Saud Medical City with a total bed capacity of 270 (241 active and 29 short-stay beds). King Saud Medical City is a large tertiary care hospital of the ministry of health in Saudi Arabia. The medical records of neonates aged 0–28 days who were treated in Neonatal Intensive Care Unit (NICU) with intravenous daptomycin as monotherapy or combination therapy for at least 4 days for proven gram-positive infection between June 2019 to July 2020 were evaluated to be included in this study. We excluded patients with incomplete data or who used daptomycin for less than 4 days. The Hospital Ethics Committee of King Saud Medical City approved (H1R1-04-Jun 20-02) the study and waived patients’ consent due to the retrospective nature of the study. 

### 4.2. Data Collection

Demographic and clinical data such as age, sex, birth weight, gestational week, chronological age, postmenstrual age, length, diagnosis for admission, dosage, frequency, and reason for daptomycin administration; concomitant administration of other antibiotics; causative organisms; susceptibility; and sensitivity tests were collected for the patients in the study using a specific data collection sheet. The, the data were transferred to an Excel sheet. Parameters and adverse events such as white blood cell, platelet, hemoglobin, calcium, corrected calcium, phosphate, INR, liver, and kidney function tests; creatine kinase (CPK), muscular pain, drug-related fever, rash, and anaphylactic reactions; eosinophilic pneumonia; and outcomes were monitored during and after discontinuation of daptomycin. Early-onset sepsis refers to the presence of sepsis within 48 h after birth and late-onset is sepsis 48 h after birth.

### 4.3. Clinical Outcome Definition:

The clinical outcome measure is classified according to the following definitions: (1) Clinical cure: The resolution of physical symptoms and signs attributed to the infectious process under daptomycin treatment based on clinical evaluation with microbiological eradication, which is defined as the absence of original bacteria in subsequent blood culture, and normalization of inflammatory markers. (2) Clinical failure: Lack of satisfying clinical response to the daptomycin according to clinical judgment, persistent bacterial growth, radiological evidence for progression of infection, potential adverse event lead to discontinued daptomycin, or death contributed to gram-positive infection while the patient on daptomycin.

### 4.4. Daptomycin Dosing

Due to the scarcity of evidence in using daptomycin for neonates, the daptomycin dose was decided by providers based on the clinical condition. Moreover, daptomycin was not stocked in the in-patient pharmacy stock in our institution. The availability of the medication was contingent upon external medication orders, and prescribing daptomycin was empirical without the availability of MIC data.

### 4.5. Data Analysis

Descriptive analysis (mean with standard deviation, median range frequencies, and percentages) were used to describe quantitative and categorical variables as appropriate. Analysis was conducted using Using R Core Team (2020) software (R Foundation for Statistical Computing, Version 4.0.1, Vienna, Austria).

## 5. Conclusions

To conclude, our study demonstrated the therapeutic benefit of daptomycin at 6 mg/kg twice daily dosing in neonates with CoNS infections who had failed the vancomycin course. Additionally, long-term administration of daptomycin did not produce any significant adverse effects that warrant its discontinuation. 

## Figures and Tables

**Table 1 antibiotics-10-00168-t001:** Baseline characteristics.

Characteristic	Total (*n* = 21)
Gestational age, median(range)	27 (24–37)
Chronological age (days), median (range)	5 (2–26)
Weight (gram), median (range)	870 (530–3700)
Female, *n* (%)	11 (52.4%)
**Mature status**	
Full term, *n* (%)	1 (4.8%)
Pre-term, *n* (%)	20 (95.2%)
**Diagnosis**	
Endovascilitis	1 (4.8%)
Late onset of sepsis	7 (33.3%)
Infective Endocarditis	9 (42.9%)
Early onset of sepsis	3 (14.3%)
Bacteremia, *n* (%)	1 (4.8%)
**Causative organism, *n* (%)**	
*Staphylococcus epidermidis*	15 (71.4%)
*Staphylococcus saprophyticis*	1 (4.8%)
*Staphylococcus haemolyticus*	3 (14.3%)
*Staphylococcus capitis*	1 (4.8%)
*Staphylococcus hominis*	1 (4.8%)
**Vancomycin MIC, *n* (%)**	
0.5 mcg/mL	2 (9.5%)
1.0 mcg/mL	10 (47.6%)
2.0 mcg/mL	6 (28.6%)
Missing	3 (14.3%)
Daptomycin Monotherapy, *n* (%) *	20 (95.2%)
Duration of therapy (days), median (range)	22 (4–43)
Duration of Follow-up (days), median (range)	82.5 (37–176)
**Dose of Daptomycin**	
10 mg/kg once daily	10 (47.6%)
6 mg/kg twice daily	8 (38.1%)
Missing data	3 (14.3%)

* Rifampicin was included in one of the cases along with daptomycin.

**Table 2 antibiotics-10-00168-t002:** Dosage, duration, and clinical outcomes.

Diagnosis, *n* (%)	Daptomycin Duration, Days, Median (Range)	Daptomycin Dosage, mg/kg/dose/day, Median (Range)	Cured, *n* (%)	Cause of Death, *n* (%) ^†^
Total, 21 (100%)	22 (4–43)	10 (6–12)	13 (61.9%)	08 (38.09) GPI: 4 GNI: 3 Abd perf: 1
Infective Endocarditis, 9 (42.9%)	12 (8–43)	10 (10–12)	5 (55.6%)	GPI: 3 GNI: 1
Late onset of sepsis, 7 (33.3%)	30 (4–42)	10 (6–12)	5 (71.4%)	GNI: 1 Abd perf: 1
Early onset of sepsis, 3 (14.3%)	12 (9–22)	10 (no range)	2 (66.7%)	GPI:1
Bacteremia, 1 (4.8%)	19 (no range)	10 (no range)	-	GNI: 1
Endovasculitis, 1 (4.8%)	30 (no range)	10 (no range)	1 (100%)	-

GPI = Gram positive infection, GNI = Gram-negative infection. Abd per: Abdominal perforation. ^†^ Percentages were calculated based on the total death number.

## Data Availability

The data of individual patient is not publicly available.

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
