# Peer review of "Safety and Efficacy of Daptomycin in Neonates with Coagulase-Negative Staphylococci: Case Series Analysis"

_antibiotics, 2021, doi:10.3390/antibiotics10020168_

Round 1

Reviewer 1 Report

This manuscript identified the efficacy and safety of daptomycin against coagulase negative staphylococci, especially in neonates. Although this study has a limitation in that the study population is small, this study is significant as a study on daptomycin conducted in neonates.

  1. In line 86 & line 81, vancomycin-resistant CNS may not be the only cause of datpomycin against CNS. In fact, it appears that no vancimycin-resistant CNS were found in this study. I suggest the background related to the introduction of daptomycin rather than vancomycin in the coagulase negative staphylococci should be added.
  2. In line 89-91, I suggest to add a reference for preclinical study on neonatal animal model.
  3. In line 105-111, I suggest to indicate numbers in addition to ratio considering that the total study population is small.
  4. In line 108, The authors described that all newborn infants who used daptomycin in this study were vancomycin treatment failure. However, considering vancomycin mic, all strains were not resistant to vancomycin. It is suggested to give the reason for vancomycin treatment failure.
  5. In line 113, This sentence seems to correspond to the method.
  6. In line 126-132, There is a question as to whether the expression higher is appropriate. Was there a control group? It would be better to substitute more appropriate expressions.
  7. In 126-127, taking into account the contents, S. hominis appears to have been given daptomycin twice a day as therapy. If so, the high clinical cure rate is likely not due to the strain factor, but due to the factor of the regimen of antibiotics.
  8. In table 2, in the cause of death, the ratio does not appear to need to be shown. Each percentage is divided into the table again, which seems to only add confusion to the reader.
  9. In 148-149, it is necessary to specifically explain whether daptomycin is better than vancomycin rather than describe "some characteristics."
  10. In 162, it seems unreasonable to judge that daptomycin is better only by clinical success. Additional basic research data such as pharmacokinetics may be needed to support this result. Moreover, the study population is too small.

Author Response

Manuscript ID: antibiotics-1088010
Type of manuscript: Article
Title: Safety and Efficacy of Daptomycin in Neonates with Coagulase-Negative
Staphylococci: Retrospective Case Series Study

This manuscript identified the efficacy and safety of daptomycin against coagulase negative staphylococci, especially in neonates. Although this study has a limitation in that the study population is small, this study is significant as a study on daptomycin conducted in neonates.

Response: Thanks for your words respected reviewer 

  1. In line 86 & line 81, vancomycin-resistant CNS may not be the only cause of datpomycin against CNS. In fact, it appears that no vancimycin-resistant CNS were found in this study. I suggest the background related to the introduction of daptomycin rather than vancomycin in the coagulase negative staphylococci should be added.

Response: The authors are thankful to the observation of the respected reviewer. We agree that there are number of causes for the use of daptomycin in addition to vancomycin resistant CNS. Background on this aspect is now added in the manuscript with references.

  1. In line 89-91, I suggest to add a reference for preclinical study on neonatal animal model.

Response: Thanks for your comment, reference is added now.

  1. In line 105-111, I suggest to indicate numbers in addition to ratio considering that the total study population is small.

Response: We appreciate reviewer comment, numbers are added in addition to ratio in the current version of manuscript.

  1. In line 108, The authors described that all newborn infants who used daptomycin in this study were vancomycin treatment failure. However, considering vancomycin mic, all strains were not resistant to vancomycin. It is suggested to give the reason for vancomycin treatment failure.

Response: The authors are thankful to respected reviewer for raising this point.

Continuous infusion of vancomycin was an option in the absence of desired therapeutic response of its intermittent dosing, however, for patients at high risk for renal impairment such as pre-term neonates as in this study, most of the neonates (20 out of 21, 95.2%) were preterm and possibly with decreased number of nephrons than full term birth neonates, availing an alternative agent such as daptomycin was considered.

A similar statement is included in the manuscript.

  1. In line 113, This sentence seems to correspond to the method.

Response: We are sorry for this mistake.

This sentence is deleted from results section and included in the methods

  1. In line 126-132, There is a question as to whether the expression higher is appropriate. Was there a control group? It would be better to substitute more appropriate expressions.

Response: Thanks for your comment. Expressions are modified now.

  1. In 126-127, taking into account the contents, S. hominis appears to have been given daptomycin twice a day as therapy. If so, the high clinical cure rate is likely not due to the strain factor, but due to the factor of the regimen of antibiotics.

Response: Thanks for your comments. The details given in this paragraph was just description of results. Yes, we agree that the antibiotic regimen has a favorable factor than the strain.

A statement on this aspect is included in results and discussion.

  1. In table 2, in the cause of death, the ratio does not appear to need to be shown. Each percentage is divided into the table again, which seems to only add confusion to the reader.

Response: Thanks, it is removed now.

  1. In 148-149, it is necessary to specifically explain whether daptomycin is better than vancomycin rather than describe "some characteristics."

Response: The authors are thankful to the reviewer for raising this issue. We have added a further explanation to strengthen the justification for the use of daptomycin.

  1. In 162, it seems unreasonable to judge that daptomycin is better only by clinical success. Additional basic research data such as pharmacokinetics may be needed to support this result. Moreover, the study population is too small.

Response: We agree with you respected reviewer. Suitable correction are done in the manuscript now to comply with your suggestion.

Reviewer 2 Report

Title:

“Safety and Efficacy of Daptomycin in Neonates with Coagulase-Negative Staphylococci: Retrospective Case Series Study”

Change “Retrospective case series study” with “case serie analysis” as it was not a study, but an analysis of clinical data in real life, and a case serie analysis in real life is observational, but neither retrospective, neither prospective.

Maybe “off-label Daptomycin” in place of “Daptomycin”: look at following paragraph.

Main point (ethical)

At first reading, I was asking myself whether such a manuscript is appropriate ethically (off-label use of “adult” antibiotic in premature neonates). After looking to reference 14 of the authors (KID list), I feel comfortable as Meyers & al. state that there is weak evidence on toxicity of Daptomycin in neonates, even if they included this molecule in their KID list (principe de précaution). So, the authors should more insist in their manuscript that even if this molecule is in the KID list, there is weak evidence, and there is a balance of risk to weight between “unknown risk in neonate of a very efficient Abiotic in adults” and “severity of CNS in premature neonates”.

It should be very useful that the authors state that the surviving children will be examined agait (for example, at 1 year) for detecting potential neuro-muscular damage by neuro-pediatric specialists (if this is possible, of course).

Abstract line 36:

There has been an increase in the prevalence of gram-positive bacteremia in neonates in the last two decades.”

Add: … as a consequence of better care and increased survival of premature neonates.

Abstract lines 48-49:

“However, no muscular 48 or neurological toxicity of daptomycin was documented in any of the cases.”

Continuing ethical discussion: there should be stated somewhere in your introduction and/or in your discussion that evidence of daptomycin (Cubicin®) toxicity in neonates and children is very week: in your reference 14, it is stated:

Daptomycin71

Neuromuscular and skeletal adverse events

Caution in <1 year

Weak

Very low

It is important to indicate that ethically, you estimate that there is no evident refusal to use this medication off label.

Introduction Lines 81-82:

Table 1 line 117: Endovasculitis

Table 1 line 117:

Daptomycin Monotherapy, n (%) 20 (95.2%): which was the combined antibiotic for the unique case not included in Monotherapy? Maybe add this information to your table as a note.

Table 1 line 117:

Vancomycin data: were all neonates treated first with vancomycin 5 days, at the dosage indicated?

If yes, please add “5 days” next to the dosage. If not, please indicate the range of vancomycin duration. This is central to your observations. Please, clarify this aspect even if you say something at lines 108-109:

“treatment failure which is defined as persistent bacterial 108 growth on top of vancomycin treatment (5 days)” In all cases ?

Author Response

Manuscript ID: antibiotics-1088010
Type of manuscript: Article
Title: Safety and Efficacy of Daptomycin in Neonates with Coagulase-Negative
Staphylococci: Retrospective Case Series Study

Change “Retrospective case series study” with “case serie analysis” as it was not a study, but an analysis of clinical data in real life, and a case serie analysis in real life is observational, but neither retrospective, neither prospective.\

Response: Thanks for your words respected reviewer. The title is changed now to "Case series analysis"

Maybe “off-label Daptomycin” in place of “Daptomycin”: look at following paragraph.

Main point (ethical)

At first reading, I was asking myself whether such a manuscript is appropriate ethically (off-label use of “adult” antibiotic in premature neonates). After looking to reference 14 of the authors (KID list), I feel comfortable as Meyers & al. state that there is weak evidence on toxicity of Daptomycin in neonates, even if they included this molecule in their KID list (principe de précaution). So, the authors should more insist in their manuscript that even if this molecule is in the KID list, there is weak evidence, and there is a balance of risk to weight between “unknown risk in neonate of a very efficient Abiotic in adults” and “severity of CNS in premature neonates”.

Response: Thanks for your suggestion. Justification of daptomycin use is now included in the introduction section. Indeed we are thankful for your useful note on the weak evidence on toxicity and the balancing statement.

It should be very useful that the authors state that the surviving children will be examined agait (for example, at 1 year) for detecting potential neuro-muscular damage by neuro-pediatric specialists (if this is possible, of course).

Response: Thanks for your suggestion respected reviewer. A statement on the re-examination of surviving children is now included in the discussion section.

Abstract line 36:

There has been an increase in the prevalence of gram-positive bacteremia in neonates in the last two decades.”

Add: … as a consequence of better care and increased survival of premature neonates.

Response:

Thanks for your suggestion respected reviewer. We agree that the 'as a consequence of better care, increased survival of premature neonates are possible. It is included now.  

Abstract lines 48-49:

“However, no muscular 48 or neurological toxicity of daptomycin was documented in any of the cases.”

Continuing ethical discussion: there should be stated somewhere in your introduction and/or in your discussion that evidence of daptomycin (Cubicin®) toxicity in neonates and children is very week: in your reference 14, it is stated:

Daptomycin71

Neuromuscular and skeletal adverse events

Caution in <1 year

Weak

Very low

It is important to indicate that ethically, you estimate that there is no evident refusal to use this medication off label.

Response: The detailed description on this issue is included in the last paragraph. We are thankful to the reviewer.

Introduction Lines 81-82:

Table 1 line 117: Endovasculitis

Table 1 line 117:

Daptomycin Monotherapy, n (%) 20 (95.2%): which was the combined antibiotic for the unique case not included in Monotherapy? Maybe add this information to your table as a note.

Response: Thanks for your comment. Information on the second antibiotic is now included as footnote of table 1

Table 1 line 117:

Vancomycin data: were all neonates treated first with vancomycin 5 days, at the dosage indicated?

Response: All cases selected in this study had vancomycin 5 days treatment before administration of daptomycin.

If yes, please add “5 days” next to the dosage. If not, please indicate the range of vancomycin duration. This is central to your observations. Please, clarify this aspect even if you say something at lines 108-109:

Response: We understand the importance of this aspect and therefore, detailed description added in section 2.1.

 “treatment failure which is defined as persistent bacterial 108 growth on top of vancomycin treatment (5 days)” In all cases ?

Response: Thanks for your comments respected reviewer. All cases included had vancomycin failure.

Round 2

Reviewer 1 Report

I would like to thank the authors for their reply to my opinion. However, the manuscript appears to have been partially modified.

  1. In the introduction, background considering daptomycin rather than vancomycin for coagulase negative staphylococci is insufficient.
  2. It is necessary to define the reason why patients in this study judged vancomycin treatment failure. Whether intermittent administration of vancomycin means sub-treatment or nephrotoxicity is not well understood. If it is a sub-treatment dose, it can be indicated by the drug level. All of the above seems to be part of the discussion.

Author Response

Response to the Comments and Suggestions of Reviewer 1 (2nd revision)

I would like to thank the authors for their reply to my opinion. However, the manuscript appears to have been partially modified.

Response: Thank you very much respected reviewer for your observation

  1. In the introduction, background considering daptomycin rather than vancomycin for coagulase negative staphylococci is insufficient.

Response: We are grateful to your comments. Additional description on the background for the use of daptomycin over vancomycin is included with two more references.

  1. It is necessary to define the reason why patients in this study judged vancomycin treatment failure. Whether intermittent administration of vancomycin means sub-treatment or nephrotoxicity is not well understood. If it is a sub-treatment dose, it can be indicated by the drug level. All of the above seems to be part of the discussion.

Response: We are indeed thankful for this comment. Now we have included the dose (15 mg/kg, twice daily) of vancomycin that was given for five days. Vancomycin treatment failure was judged after finding persistent bacterial growth at the end of five days of vancomycin treatment. Now intermittent dosing corrected to twice daily dosing.
